# The Forgotten Brother: The Innate-like B1 Cell in Multiple Sclerosis

**DOI:** 10.3390/biomedicines10030606

**Published:** 2022-03-04

**Authors:** Saar T. Halperin, Bert A. ’t Hart, Antonio Luchicchi, Geert J. Schenk

**Affiliations:** Department of Anatomy and Neurosciences, Amsterdam Neuroscience, MS Center Amsterdam, Amsterdam UMC, Vrije Universiteit, 1081 HZ Amsterdam, The Netherlands; s.halperin@amsterdamumc.nl (S.T.H.); mog3556@gmail.com (B.A.’t.H.)

**Keywords:** multiple sclerosis, B1 cell, autoimmunity, autoantibodies, innate-like lymphocytes, natural antibodies, neurodegeneration

## Abstract

Multiple sclerosis (MS) is a neurodegenerative disease of the central nervous system (CNS), traditionally considered a chronic autoimmune attack against the insulating myelin sheaths around axons. However, the exact etiology has not been identified and is likely multi-factorial. Recently, evidence has been accumulating that implies that autoimmune processes underlying MS may, in fact, be triggered by pathological processes initiated within the CNS. This review focuses on a relatively unexplored immune cell—the “innate-like” B1 lymphocyte. The B1 cell is a primary-natural-antibody- and anti-inflammatory-cytokine-producing cell present in the healthy brain. It has been recently shown that its frequency and function may differ between MS patients and healthy controls, but its exact involvement in the MS pathogenic process remains obscure. In this review, we propose that this enigmatic cell may play a more prominent role in MS pathology than ever imagined. We aim to shed light on the human B1 cell in health and disease, and how dysregulation in its delicate homeostatic role could impact MS. Furthermore, novel therapeutic avenues to restore B1 cells’ beneficial functions will be proposed.

## 1. Introduction

### 1.1. Multiple Sclerosis and B1 Cells

Multiple sclerosis (MS) is a chronic neuro–immunological demyelinating disease of the central nervous system (CNS) characterized by prototypic multifocal neurodegeneration of white and gray matter [1]. The number of MS patients is rising globally and was estimated in 2016 at around 2.2 million. Western European and high-income North American countries currently have the highest prevalence and a significantly increasing rate (up 30% from 1990) of MS patients [2]. Most patients are young adults, with a mean disease onset age of 30 years, and women on average are three times more often affected than men [3,4]. A large arsenal of new therapeutics has been developed in recent years that delay MS progression and reduce the chance of disability [5,6]. These new-generation medicines have helped improve the quality of life and extend the life expectancy of patients; however, MS still is uncurable and forms a substantial burden on patients, families, and the health care system [2].

Recent biotechnological advances in molecular, cellular, and imaging techniques have improved understanding of the biological processes that govern MS. However, the exact etiology and mechanisms involved in its pathogenesis are still not fully understood. Nevertheless, a growing body of evidence points out that MS is of heterogeneous origin [1]. Over the years, it was proposed that MS could be caused by a multifactorial combination of genetic predisposition, gender, age, geographic location, polymorphisms in immune-related gene expression, exposure to pathogens, or nutrition [7,8,9].

The mounting recognition that MS pathophysiology is related to neuroinflammatory processes has nourished translational and clinical research lines in the field and yielded important discoveries. For example, (1) at the translational level—the presence of autoantibodies (aAbs) against myelin or damaged neuronal derivates (compromised self-antigens; neoantigens)—was confirmed in the blood plasma and the cerebral spinal fluid (CSF) of MS patients [10,11]. Indeed, a fluctuation in these aAb levels is considered a potential biomarker for the diagnosis of MS [12,13]. (2) On the other hand—at the clinical level—the success of new B lymphocyte (B cells) targeted therapies led to the improvement in clinical symptomology and underscored the strong neuroinflammatory link to MS pathophysiology [14]. As a result, neuroimmunologists began to shift their attention toward B cells’ involvement in the immunopathology of MS and suggested that their role could span beyond mere antibody secretion and long-term immune memory. B cells participate in different stages of MS; they take part in antigen presentation to self-reactive T cells. Thereby they might trigger an (unsought) immune response and can secrete pro- and anti-inflammatory cytokines (a group of immunomodulatory signaling proteins) [15,16,17].

This review focuses on the role of a unique B-cell subtype that participates in innate immunity, contributes to autoimmunity, and is presumably also involved in MS. The “innate-like B1 cell” (B1 cell) was first described in mice as a “non-conventional B lymphocyte” in the 1980s [18,19]. However, its human counterpart has only recently been confirmed in peripheral blood [20,21,22,23,24]. As B1 cells’ functional importance is just starting to unravel, this review’s primary goal is two-fold: (1) to provide an up-to-date overview of the human B1 cell’s physiological role in health and (2) to discuss evidence for its role in MS and the consequence of its dysregulated physiology.

### 1.2. B1 Cell—A Unique B-Cell Subset

B1 cells belong to a unique subpopulation of “lymphocyte-like” cells that are phenotypically related but functionally distinct from the traditional B lymphocytes, denoted as B2 cells [25]. B1 cells are critical players in the first line of defense against pathogens or self-antigens from damaged tissues [26,27,28]. Even though B1 cells constitute a small percentage of the adult leukocyte pool in adult humans (1–3% of total adult B cells in the blood and peritoneal cavity) [22,29,30], they are nevertheless responsible for the constitutive production of natural antibodies (nAbs) and secretion of immunomodulatory molecules [26,31,32]. Antibodies generated by B1 cells bind to self-antigens, such as damaged myelin-debris and apoptotic cells [33,34]. Furthermore, studies in mice showed that they might also be involved in the protection and myelination of axons during brain development [15,35] or after brain injury [36]. Therefore, it has been suggested that B1 cells have evolved to fulfill a housekeeping role, to regulate inflammation, and to promote tolerance to “self” [37]. Altogether, B1 cells might play a more prominent role in autoimmunity and MS than ever presumed.

At the time of their discovery (in the 1960s), “conventional” B cells were categorically considered to consist of two types of cells: plasma and memory B cells [38]. The first was considered a short-lived, fully differentiated B lymphocyte that produces a high amount of antigen-specific antibodies against pathogenic foreign or damaged-self byproducts. The latter was seen as a long-lived memory cell that recognizes antigens the body had previously been exposed to and launches a faster, more efficient immune response upon re-exposure [38]. Given these properties, both B cell types were considered to belong to the immune response’s adaptive arm. However, it is currently well acknowledged that the division might be too simplistic and not accurately represent B cells’ convolution and functional repertoire [39,40]. In recent years, the B cell ontogenetic diversity was revealed to consist of distinct subpopulations of cells. While the “conventional” B cell (B2 cells) forms the largest B-cell population in the blood [41,42], the B1-cell subtype is a small, diverse, and dynamic subpopulation that belongs to the innate immune arm [24] based on characteristics we will deliberate on below. This evolutionarily conserved phagocytic B cell is present as early as in teleost fish [43,44] and has distinctive developmental, morphological, and functional properties that lucidly distinguish it from other lymphocytes [21,25,45].

B1 cells were first described in mice by Hayakawa et al., as a unique population of splenic B cells with specific properties, such as their ability to “naturally” secrete immunoglobulins of class type M (IgMs)—in a T-cell-independent fashion [18]. B1 cells were named so because they are the first B lymphocytes to develop (in the embryonal fetal liver and subsequently in the neonatal bone marrow) and because of their presence from early in the evolution of lower organisms [46]. These properties imply that B1 cells are an old relic of immune cells that served an innate immune function until B2 cells’ appearance. However, as we will show in this review, the evidence is accumulating that B1 cells are more than a merely primitive lymphocyte subpopulation. On the contrary, it has been demonstrated that B1 cells hold a unique position: they play a pivotal role in bridging the innate and adaptive immune response [47,48]. Additionally, B1 cells fulfill distinctive, stand-alone functions promoting homeostasis and safeguarding immune tolerance [49,50,51]. 

Since their first identification in mice, many studies about B1 cells’ characteristics and functions have been amassing. However, proof of its existence in humans was lacking for a long time [52]. As a critical discussion of mouse B1 cells is outside the scope of the manuscript (MS is a human disease without a proper rodent equivalent), we refer the reader to some excellent reviews on the basic immunobiology of mouse B1 cells [31,32,37,53,54].

Confirming the presence of a “mice-equivalent” B1 cell in humans was a challenging task since no exclusive phenotypical and functional characterization biomarkers for B1 cells were available [23]. Moreover, early studies on human B1 cells employed less-sensitive or inconsistent research methodologies showing varying results (several excellent papers summarize these findings [21,23,24,45,47,55,56,57,58]. However, recent use of advanced technologies and innovative methodologies yielded evidence for specific surface-marker expression profiles unique to human B1 cells [21,52,59]. It appears now that some B1 cells acquire phenotypical properties that traditionally characterize myeloid cells, such as macrophages [60,61,62,63]. Nevertheless, these B1 cells might retain, besides their surface-expressed IgM receptors, the unique cell-surface expression markers of B2 cells (e.g., CD19) [24,55,64]. Recently, immunohistochemical criteria for identifying human B1 cells were proposed, and their implementation in current studies is growing in acceptance [21,22,24,25,29,30,64,65,66,67,68,69,70,71] (Figure 1).

### 1.3. B1-Cell Subsets Have Distinctive Phenotypes and Functions

Substantial evidence supports the idea that the B1-cell population is not as homogenous as previously thought [31,60,77,80,81,82,83,84,85]. Some B1-cell types express distinctive surface markers. For example, at least two distinct subpopulations are recognized in mice: CD5^+^ B1 cells (B1a cells) and CD5^−^ B1 cells (B1b cells) [86]. The phenotypical differences between the two subsets are attributed to their specific surface markers, functionality, and tissue-distribution variation [25,87,88].

For example, it is now well-accepted that B1a cells produce most of the natural IgM Abs in the serum and participate in the first line of defense, whereas B1b cells are essential for developing IgM memory B1 cells and form a bridge to the adaptive immune response [34,79]. Supporting evidence comes from adoptive transfer experiments in mice showing B1b cells mediated long-lasting IgM memory to infection [89]. B1b cells were shown to fulfill a more “APC-like” (antigen-presenting cell) function than B1a cells. Upon activation, B1b cells migrate to secondary lymph organs and differentiate into Ab-producing plasma cells [90]. They undergo somatic hypermutation and affinity maturation, increasing their specificity [36,37,90,91,92]. Notably, this all occurs in the absence of T-cell help. 

Besides B1a and B1b cells, recent studies identified other B1 cell subsets in mice and humans based on phenotypical and functional analysis. Different authors proposed that each B1-cell subset may contribute differently to the immune response and have different clinical implications in autoimmune diseases. For example, the B1-cell-derived phagocyte (B1-CDP) is a B1a-cell-derived subset identified in mice that can migrate to inflammatory sites in the tissue [53,83,93]; after differentiation, it acquires morphological and functional characteristics similar to those of phagocytes [60,94,95,96] (nevertheless, it maintains a B1-cell’s lymphoid-marker characteristics [97,98]. Another example is the innate-response-activator B1 cell (IRA-B1), which produces the pro-inflammatory cytokines GM-CSF and IL-3 [99] and stimulates B1a cells to secrete intracellular nAbs stocks [83,87,100,101]. Notably, B2 cells do not produce GM-CSF [73,102]. The proposed main B1-cell subgroups are depicted in Figure 1B [60,65,68,78,103,104,105,106,107,108]. However, it remains to be established whether these B1-cell subtypes in humans are committed subsets or adaptations of a multifunctional B1-precursor cell to the tissue environment. 

## 2. B1 Cell’s Functions in Health

### 2.1. First Responder to Danger

Once activated, mice B1 cells move from their primary residence in the peritoneal/pleural cavities or the spleen [31,101,109,110,111,112,113]—and presumably—from the circulation in humans [30,114] to secondary lymphoid organs draining the compromised tissue. Hence, it was suggested that B1 cells might represent a reservoir of innate lymphocyte armor that can speedily relocate to the tissue in response to homeostatic disturbance [49]. Indeed, it was recently shown that human B1 cells readily migrate to inflammatory [115] or wound-healing lesions [97]. Once onsite, they begin to secrete IgMs and cytokines rapidly [116,117,118,119]. However, the exact pool location of B1 cells in humans remains to be identified.

### 2.2. Phagocytosis—Eating for Elimination

Phagocytosis is an evolutionarily preserved immune mechanism that provides adequate first-line protection against pathogenic microorganisms and autoantigens [120]. Phagocytosis of large particles is executed predominantly by specialized phagocytes such as neutrophils, dendritic cells, or macrophages [121]. Until recently, it was believed that naïve B2 cells, let alone B1 cells, could not perform phagocytosis of large particles (>0.5 µm) because of their smaller size, lack of motility, and absence of a phagosome compartment or scavenger receptors [43,122]. However, it has been recently demonstrated that murine and human B1 cells can phagocytose large particles utilizing pseudopodia-like cytoplasmic extensions [62,123]. Several studies confirmed that murine B1-CDP cells could phagocytose solid particles, such as latex beads and microbes, in vivo. Interestingly, B1-CDP cells had a significantly higher phagocytic activity than peritoneal macrophages [94,95,96,124,125,126,127]. 

### 2.3. Antigen Presentation—Eating for Alerting Others

Besides natural-antibody secretion and debris removal, B1 cells contribute to immunity by other means—both in a stimulatory and suppressive manner. B1 cells directly induce activation of CD4^+^ T cells by ovalbumin (OVA) or dinitrophenol-keyholc limpet hemocyanin (DNP-KLH) from the periphery [62,128,129,130] or after exposure to influenza virus [131], bacteria, fungi, or parasites [118,132]. In line with these observations, other studies have demonstrated that B1 cells are efficient APCs, comparable to B2 cells or dendritic cells [34,62,130,133]. In relevance to autoimmunity, B1a cells were shown to act as highly potent APCs for autoantigens such as double-stranded DNA [133]. Interestingly, in mice and a lupus-prone mouse model, B1a cells presenting OVA peptides in vitro induced twice the levels of T-cell-derived INF γ, IL−4, and IL−10 than B2 lymphocytes [80,128,129,130,134]. Furthermore, several studies have shown that *activated* B1 cells express higher levels of MHC class II and the costimulatory molecules CD80, CD86, and CD40 [60,133,135]. The role of B1 cells as a potent APC was recently reviewed by Popi et al. [34]. 

### 2.4. Humoral Response—Arrows to the Target

In contrast to the cell-mediated cytotoxic response (e.g., macrophages) that directly eliminates pathogens, humoral immunity involves defense mechanisms mediated by secreted molecules into the extracellular environment [136]. Cytokines are an important example of soluble effector molecules that facilitate intercellular communication between (immune) cells. Cytokines are produced inside the CNS by immune cells, including B1 cells, to modulate responsiveness, differentiation, or maturation of immune cells and contribute to their migration to and throughout the CNS (reviewed in [137,138,139]). Chemokines (chemoattractant cytokines) can be conveyed into the brain by selective transporters or produced locally within the brain by infiltrating or by resident immune cells, endothelial cells, or even nerve and glial cells [140,141,142].

Furthermore, a large body of evidence shows that B1 cells adopt an immune regulatory function. Large amounts of the cytokine IL-10 are constitutively secreted by B1 cells to dampen inflammation and promote homeostasis [49,86]. IL-10 was shown to promote homeostasis by alleviating inflammatory responses and inducing immune tolerance in patients with autoimmune diseases [51]. Moreover, IL-10 can inhibit antigen presentation—and hence T-cell proliferation [111,143,144]. Interestingly, it was shown that peritoneal mouse B1 cells secrete more IL-10 than B2 cells without antigen exposure or T-cell activation and can even increase its production after antigenic challenge [49,86,145]. Recently, Aziz et al., observed that mice, adoptively transferred with B-1a cells, showed dramatic improvement in lung injury and apoptosis by reducing the production of pro-inflammatory cytokines (e.g., IL-6 and IL-1β) [146].

Moreover, it was recently shown that the migration of B1 cells to the spleen after parasitic infection increases the percentage of T regulatory cells (Tregs). This finding entails B1-cells’ participation in Tregs activation during infection [132]. Recently, Hsu and colleagues designated a group of B1a cells in mice, termed ‘Treg-of-B1a’ cells [82]. These cells were able to convert naïve T cells into a novel subset of anti-inflammatory Tregs in an IL-10- or TGF-β-independent manner [82]. Besides, B1 cells have a suppressive effect on macrophages beyond a mere inhibition of their phagocytic ability. In vitro experiments (in mice) showed that the release of reactive oxygen species by macrophages was impaired in the presence of B1 cells [147]. Impairing macrophages’ aptitude to eliminate damaged self-antigens and restore homeostasis could sustain an adverse pro-inflammatory reaction in the case of MS. In this scenario, degraded-myelin debris might not be removed effectively and, as a result, accumulate in the proximity of injured axons.

Intriguingly, B1 cells also produce pro-inflammatory cytokines and take on a pro-inflammatory role under certain circumstances (e.g., aging or under immunogenic stimuli) [51,65,148,149]. For example, adoptive transfer experiments in parasite-infected mice showed that B1 cells produce IFN-γ, TNF-⍺, IL-2, and IL-4—essential activators of cellular and humoral T-cell responses [150]. Another piece of evidence for the contribution of B-1 cells to autoimmune pathogenesis arises from experiments in the lupus mouse model. For example, deleting peritoneal B-1 cells by hypotonic shock reduced disease severity in mice [32,151]. Moreover, it was previously reported that in mice, a novel B1 cell subpopulation bearing programmed-death-ligand 2 (PDL-2) could stimulate Th17 cell differentiation, thereby favoring a pro-inflammatory response (as opposed to B2 cells in the same experiment) [152]. These cells preferentially switched immunoglobulin isotypes to IgG1 and IgG2b in the presence of IL-21 (a pro-inflammatory cytokine involved in activating CD8 T cells during chronic infections) [152,153].

Furthermore, in contrast to the prevailing paradigm (i.e., IL-10 is a “classic” anti-inflammatory cytokine), several reports show that IL-10 might also have a pro-inflammatory mode of action, depending on the inflammatory context. For example, IL-10 increased NK cells’ cytotoxicity or produced IFNγ, instigating immune activation [154,155,156]. On the other hand, other studies [157,158] have shown that B1-cell-derived IL-10 has an anti-inflammatory effect, for example, in MOG-peptide-induced EAE [157]. Because the B1 cell is a major source of IL-10 [159,160], it is, therefore, possible that under certain circumstances (i.e., before an MS relapse) B1 increases its production and attenuates its anti-inflammatory properties (Figure 5). 

### 2.5. Natural Autoantibodies

B1 cells organically produce antibodies against compromised self-antigens even without antigenic stimulation (i.e., under steady-state conditions) [161]. These antibodies are referred to as “natural-antibodies” (nAbs). They facilitate debris clearance by opsonizing debris, activating the complement system, and communicating with other innate immune cells [64,162]. The logic behind the terminology is that they are innately programmed to exert immune-protective roles as a “ready-made” arsenal. Remarkably, B1 cells produce about 80–90% of the natural IgM and 50% of the natural IgA pool [161,163,164,165,166]. After secretion, low-affinity IgM nAbs assume a pentamer configuration that increases the strength of binding (increased avidity) [167,168]. The nAbs of the IgM isotype are polyreactive and can bind a rainbow of evolutionarily highly conserved “self” and “non-self” molecular patterns on antigens. For example, nIgM can recognize epitopes on phosphorylcholine (PC), which is present on all cell membranes of apoptotic cells and degraded myelin. Although most nAbs are of the IgM class, they can also be IgG or IgA [163,165,169,170]—a fact that might indicate a tissue-tailored activity [32,171]. Single-cell mRNA sequencing analysis from SARS-CoV-2-infected patients showed that the percentage of B1-like cells decreased in the serum. The reduction was correlated with decreased levels of serum IgM and IgD. These studies may suggest a deficit in either functionality or frequency of nAb-secreting cells in individuals infected with SARS-CoV-2 virus [172,173].

### 2.6. Myelination and Remyelination

It has been long known that B1 cells in mice are predominantly generated in the fetal liver during fetal and neonatal development [174]. However, their presence in the adult brain had not been elucidated until recently [15,35]. A well-designed study demonstrated that mice B1a cells infiltrate the neonatal mouse brain and promote the proliferation of oligodendrocyte precursor cells (OPCs) in vitro [35]. At least 50% of the B cells in the developing brain of mice were positive for B1a cell surface markers (CD19^+^, CD5^+^, CD43^+^, CD93^+^, and IgM^+^). The authors reported that nAbs, especially of the IgM isotype secreted by B1a cells, played an essential role in inducing oligodendrocyte development and maintaining immune homeostasis [15,35,175]. B1a cells accumulate in the meningeal space throughout the CNS, choroid plexus, and lateral ventricles and are involved in the development of the neonatal mouse brain [35]. Whether this is the case also with human B1a-like cells awaits confirmation. Furthermore, it is known that OPCs express receptors that recognize Fc regions of IgGs and IgMs [15,176]. So a scenario in which signaling through the IgG-Fc receptor could promote oligodendrocyte differentiation is possible [35,175]. The B1 cell’s functions relevant to MS are summarized in Figure 2.

## 3. B1 Cells in MS

### 3.1. B1 Cells Are Autoreactive by Nature

B1 cells have evolved to fulfill housekeeping roles and promote tolerance to “self” [37]. Therefore, B1 cells are, paradoxically, positively selected for their ability to recognize self-antigens, whereas B2 and T cells that recognize self-antigens with higher affinity are eliminated by negative selection [177,178]. Alternatively, it is plausible that B1 cells might escape negative selection because they mature outside the bone marrow or due to their low antigen affinity. Altogether, it is conceivable that B1 cells play a prominent role in the induction of tolerance, and dysregulation of their tolerogenic function could adversely result in the production of autoreactive antibodies (autoantibodies; aAbs) directed against “self” [48,179]. 

Approximately 2–3% of healthy humans carry natural autoantibodies (nAbs) against self-antigens in the CNS [13,180]. These nAbs can enter the brain tissue but are usually not immune-competent because their target epitopes are “hidden” [13,180,181]. However, under certain circumstances they can encounter and attack tissue antigens, such as myelin [182]. In MS, stressed axons undergo oxidative reactions, instigating “virtual hypoxia” that damages the membrane’s integrity and destabilizes myelin [183]. Similarly, post-translational modifications of myelin’s lipoproteins, such as by citrullination (discussed below), could expose self-antigens or epitopes (that would usually be masked or unreachable to nAbs)—and thereby promote inflammation [184]. More about the pathogenic features of aAbs is reviewed in reference [163]. 

B1 cell’s production of aAbs is essential for removing senescent cells, tumor cells, and cellular debris. Moreover, nAbs can neutralize or opsonize autoantigens, thereby alleviating the immunogenic burden [13,185]. Studies have shown that oligodendrocyte-reactive aAbs promote remyelination in MS and murine models [168,186,187,188,189,190]. Researchers point out that these auto-reactive antibodies could also play a vital role in fostering remyelination and axonal growth in humans [171,191,192]. However, the impaired capability of B1 cells to self-inhibit their signaling through inhibitory receptors (e.g., SIRP-alpha or CD5) could turn disadvantageous and result in physiological dysfunction, causing a lower activation threshold—and thereby over-reactivity to “self” antigens [178]. As a result, B1 cells that usually produce harmless, low-affinity, polyreactive nAbs could be activated by autoreactive T helper cells and enter or proliferate in germinal centers in the inflamed meninges [193]. There they might undergo class switching, somatic hypermutation, and affinity mutation. Subsequently, they might mature into autoreactive “pathogenic” plasma cells that constitutively produce high-affinity IgM or IgG aAbs directed against self-peptides [185,194,195]. Indeed, studies have shown a clear correlation between the percentage of B1a cells and the IgMs’ frequency inside the CNS (as measured in the CSF). This observation could imply that B1 cells are involved in intrathecal IgM production [196,197]. Moreover, high-affinity, pro-inflammatory-IgMs directed against compromised myelin lipids secreted by B1a cells have been established and associated with an aggressive MS course [196]. Along the same line of evidence, eliminating B1 cells in a lupus mouse model reduced autoimmunity and the severity the disease [32,151]. 

It was previously hypothesized that B1b cells play an active role in modulating immunological responses during MS and thus could contribute to MS pathogenesis [51]. For example, B1b cells could differentiate into plasma- and memory-B1 cells and produce increased-affinity antibodies against myelin-derived components [51]. By this, B1b could act as a double-edged sword; it can help alleviate relapse episodes and suppress progression to SPMS. On the other hand, its constitutive (over)production of such high-affinity antibodies could cause chronic immune activation and focal autoimmune attacks against otherwise physiologically healthy myelin antigens [51,92,166,198].

### 3.2. B1 Cells Are Linked to Autoimmunity

Since clonally expanded autoreactive B1 cells with somatically hypermutated BCR specificities are present in the CSF and the periphery of MS patients, it was assumed that there is a bidirectional exchange across the BBB [13,196,199]. B1 cells might identify their specific antigens within the CNS and migrate back and forth to the regional lymph nodes through the neuro–lymphatic system (similar to B2 cells) (Figure 3). This could imply that the origin of the damage is within the CNS. Whether B1 cells’ maturation into pathogenic aAbs-secreting cells also occurs in tertiary lymphoid organs, for example, in the meninges—a possibility that would aggravate pathogenesis—needs to be confirmed. Nevertheless, Villar et al., showed evidence that CD5+ (innate) B cells are the source of persistent anti-lipid IgM antibodies that constitute part of the oligoclonal IgM bands in the CSF [200]—a characteristic biomarker seen in the CSF of MS patients [117,200]. Researchers proposed that these OCBs may also originate from within the CNS by resident B1a cells [35]. The pathological significance of B1 cells in MS is reinforced by data showing that elevated levels of peripheral B1a cells and IgM OCB (presumably from CD5+ B cells) are highly predictive of poor prognosis and transition from “clinically isolated syndrome” to RRMS (CIS: a first, single episode of MS relapse-like symptoms that often progress to RRMS) [33,193,197].

### 3.3. MS-Related Autoantibody Targets in the CNS

An extensive search for antigen specificities against which aAbs are directed in MS has risen recently. Among the known aAbs produced by B1 cells that recognize self-antigens are myelin-derived components, such as phosphatidylcholine (PtC), DNA-derived antigens, or neural-derived protein aggregates such as tau and amyloid-beta [64,184,185,201,202,203,204,205]. However, a causative relationship or a direct link between a specific antigen and MS is still lacking. Figure 4 illustrates CNS-related antigens and autoantibodies that may be involved in the pathogenesis of MS. Here, we highlight three examples:

#### 3.3.1. Anti-Phosphatidylcholine aAbs

Myelin is pivotal for the proper functioning of the nervous system [206]. The myelin sheath comprises a high amount of lipids (70%) and a relatively low amount of proteins (30%) [207]. Phospholipids, such as PtC, constitute more than 12% of the total myelin lipids and are essential for the initiation, compaction, and configuration of the plasma membrane [208]. It has been reported that in the CSF of MS patients, anti-PtC IgM antibodies are common and seem to be associated with disease progression [197]. Elvington et al., showed that anti-PtC IgMs play a crucial role in activating complement and driving cerebral injury after ischemic stroke [209]. Furthermore, very recently, in a large cohort study, Sádaba and colleagues examined anti-PtC IgM levels in peripheral blood of MS patients and reported these aAbs were intensely increased during CIS and RRMS. The authors suggested that serum PtC IgMs could be a candidate biomarker for the early inflammatory stages of MS [11]. Nonetheless, one should realize that anti-PtC IgMs are also present in healthy humans’ serum, but they do not ligate non-oxidized forms of PtC. In other words, anti-PtC IgMs usually do not bind to healthy PtC [209]. This notion underscores anti-PtC IgMs’ protective role in promoting efficient clearance of myelin-debris and apoptotic cells. Anti-PtC IgM elevation in CSF of MS patients might mirror an increase in B1 cell proliferation within the CNS that, in turn, contributes to MS pathology [210].

#### 3.3.2. Citrullinated Myelin-Derived Proteins

Citrulline is a non-essential amino acid that humans cannot synthesize but can generate by post-translational modification of proteins [211,212]. Citrullination is a natural enzymatic process catalyzed by a group of five peptidyl-arginine deiminases (PADs) by which positively charged arginine residues are converted to a neutrally charged amino acid: citrulline. Consequently, the protein’s conformation changes as well as its properties and functionalities [213]. Hence, destabilization and degradation of myelin can occur—the pathophysiological hallmark of MS. Citrullinated self-proteins elicit autoreactive T-cell activation in patients with MS, RA, and other autoimmune diseases [214,215]. Importantly, citrullination can create new T-cell epitopes by altering antigen processing, as was shown for the MS-relevant autoantigen myelin oligodendrocyte glycoprotein [216]. The significance of elevated citrulline levels in the plasma of MS patients is intriguing because no direct correlation with clinical disability status or with lesion load has been established yet [217].

On the other hand, it was shown that citrullination is elevated in active and chronic lesions, and that citrullination of myelin-derived proteins precedes demyelination [218]. Furthermore, it was speculated that the brains of patients with SPMS might contain an immature form of citrullinated proteins that could predispose these individuals to an autoimmune attack [219]. In addition, there is evidence that increased amounts of citrullinated myelin are elevated earlier in the disease and could later serve as an antigenic substrate in predisposed individuals [220]. An early study showed a correlation between peripheral CD5+ lymphocytes and circulating anti-myelin basic protein (MBP) aAbs in human patients with (active) MS, especially with the disease duration and the number of gadolinium-enhancing lesions [221]. As evidence, in RRMS patients, citrullinated MBP (citMBP) levels elevate and account for about 45% of the total MBP in the brain [222]. In some severe forms of MS (Marburg’s variant), almost 100% of the MBP is citrullinated [219]. Taken together, it has been suggested that increased citrulline levels might be a promising biomarker in MS [217].

#### 3.3.3. B1-Cell-Derived Abs against Myelin

Recent studies exposed a family of natural human IgMs that attach to neural and myelin antigens. Two prominent examples are r*HIgM12* and r*HIgM22* monoclonal antibodies [187,223]; rHIgM22 is currently under investigation as a potential therapeutic agent for MS [224]. This serum-derived human nAb directly binds to (an) unknown antigen(s) on mature oligodendrocytes and promotes remyelination in Theiler’s murine encephalomyelitis virus model of MS (TMEV) [180,225]. Some indications imply that hIgM22 binds to sulfatide, an essential myelin component [182].

It was also shown that hIgM22 could bind unfixed slices of mouse cerebellum [187]. More recently, Cui et al., showed that treatment with rHIgM22 significantly accelerated hippocampal remyelination and improved hippocampal-dependent memory deficits in mice. In this study, rHIgM22 facilitated remyelination in the corpus callosum [223,226]. In another recent in vitro study, rHIgM22 was found to bind CNS-derived myelin and promote phagocytic clearance of myelin debris by microglia [227]. Therefore, it was proposed that rHIgM22 could bind myelin deposits, tagging them for phagocytic clearance. Zorina’s group concluded that microglia might play a key role in rHIgM22-induced remyelination by phagocytosing rHIgM22-bound myelin. Furthermore, they proposed that this mechanism might allow OPC differentiation in MS patients’ demyelinated lesions and stimulate remyelination [227].

MRI studies in EAE animal models showed that IgM mAbs could bind to oligodendrocytes in demyelinating lesions [225]. Mice treated with the remyelination-promoting nAb rHIgM22 significantly decreased the volume of lesions in the spinal cord [225]. Rodrigues et al., suggested a mechanism of action for the rHIgM22 IgMs. They proposed a model in which a large complex of pentameric IgM nAbs binds to several unique epitopes on the surface of myelinating cells with high avidity. This causes epitopes that normally do not interact to form a signaling complex. The result is a downstream signal to the nucleus that promotes remyelination [182]. Four weeks after treating mice with serum rHIgM22, virtually all MBP-positive oligodendrocytes bound serum rHIgM22, showing this antibody’s specificity for live oligodendrocytes. These promising results were later translated into clinical trials. A phase 1, multicenter, double-blind, placebo-controlled study was completed in 2017 and showed that rHIgM22 was well-tolerated [9,228]. Another safety and tolerability study in mice demonstrated that a single dose of rHIgM22 treatment, albeit in a low sample size, could promote remyelination in the brainstem [229]. 

Notably, an ongoing clinical study showed that despite an intact BBB, rHIgM22 molecules could penetrate the CNS of people with MS immediately after a relapse (clinical trial ID: NCT02398461). This clinical study may determine whether the BBB is more permeable to rHIgM22 in MS than healthy controls and whether the antibody can be associated with clinical or radiological findings.

Another example of a human monoclonal nAb recently found to protect exposed axons (i.e., in the absence of remyelination) is r*HIgM12*. A study in mice with TMEV-induced demyelination reported that rHIgM12, bound to neuronal surfaces, supported substantial neurite elongation [187,188]. A follow-up study confirmed that rHIgM12 is attached to glycoprotein and glycolipid antigens expressing gangliosides on axons [188,230,231]. Therefore, rHIgM12 might be promising as a therapeutic against axonal degeneration in MS.

### 3.4. B1-Cell Frequency Correlates with Relapse and MS Progression

Considering the essential homeostatic and tolerogenic functions of B1 cells, one can speculate that a dysregulation in the quantity of B1 cells may result in either excessive or impaired responses to self-antigens, leading to autoimmune responses. Indeed, aberrant numbers of B1 cells in humans have been reported in patients with various autoimmune-related diseases, such as MS [196,198,232], Sjögren’s syndrome [233], autoimmune lepromatous leprosy [234], rheumatoid arthritis (RA) [235,236], and systemic lupus erythematosus (SLE) [32,77,237]. For example, Seidi and colleagues [221] showed that an increased CD5+ B-cell percentage in peripheral blood in MS was positively correlated with gadolinium-contrast-enhancing MRI lesion load (r = 0.31; *p* < 0.05). They also showed that peripheral blood CD5+ B cells were significantly increased in patients in relapse compared to patients in remission (~15% vs. ~5%). However, the total levels of peripheral B cells (B2 plus all B1-cell subsets) or T cells in RRMS patients during relapse were similar to the control groups [221]. In contrast, a more recent study by Tørring et al., showed that the frequency of B1 cells in RRMS patients was lower than in healthy controls (~1.2% vs. ~1.7%) and inversely correlated to the time-lapse since the last attack and disease progression (r = −0.49; *p* = 0.01) [22]. Finally, the authors reported that peripheral-blood human B1 cells decreased from about 1.5% of the total B cells 2 months after the attack to less than 1% around 10 months after the attack [22]. It was suggested that this might result from B1-cell involvement in the brain’s inflammatory processes just before a relapse—or, otherwise, as a result of the requirement to dampen inflammation by vigorous production of IL-10 [22].

Tørring et al., speculated that low levels of B1 cells seen in RRMS patients’ peripheral blood could be considered a sign of immune dysregulation [22]. The mechanisms through which this is executed remain to be discovered. However, it was suggested that because some B1 cells may express higher CD11b levels than naïve B2 cells, it is conceivable that B1 cells migrate to the brain in RRMS more readily than B2 cells and, as a consequence, reduce their quantity in the peripheral blood [22]. It is worth noting that some studies have reported that, in contrast to the lower frequency in the circulation, some MS patients had elevated percentages of B1a cells in CSF compared with other neurological diseases [199].

### 3.5. MS Progression Is Age-Related; So Is B1 Cells’ Quality and Frequency

Many neurodegenerative disorders progress with age, implying a long prodromal period of pathological events that gradually manifest into clinical disease stages. Progressive MS is an example of such age-related neurological deterioration [238,239]. The aging immune system’s change is characterized by an imbalanced inflammatory response that could lead to greater susceptibility to infectious and chronic diseases [240,241]. Several reports have shown fluctuations in the B2-lymphocyte population during aging, both in mice and humans. For example, some individuals had a prominent decrease in their B-cell amount and repertoire diversity. Notably, the percentage and the absolute number of B2 cells declined in an age-dependent manner [238,241,242]. Moreover, a statistically significant age-related rise in the serum level of immunoglobulin of classes IgG and IgA but not IgM was signaled, a fact implying a shift into a more pathogenic aAb repertoire in the elderly [243] (Figure 5).

Clinical characterization of human B1 cells has not been extensively studied in the elderly, let alone longitudinally in aging MS patients. However, a recent study that analyzed the change in human B1-cell function and repertoire with age showed functional impairments in the B1-cell population with advancing age, as the B1-cell population from older individuals encompassed fewer antibody-secreting cells than younger individuals. The study also confirmed that the frequency of the B1 cells significantly decreased with advancing age [29]. The authors estimated that the human B1-cell component at the age of 30 or younger is ~ 2–3% of total CD19+ B cells and decreases to 1% above the age of 50. The frequency of human B1 cells is inversely correlated with age (B-1 cells: r = −0.34; *p* = 0.001) [29].

Furthermore, another group recently reported that B1a cells could give rise to BL4 cells—a novel subset of “aggressive” innate B1 cells [65]. The same group showed that aging influences these B1a-derived cells to lose their immune-suppressive function and become inducers of cytotoxic CD8+ T cells [65,244]. This is in line with the evident upregulation and high expression of surface molecules on BL4 cells needed for the CD8+ T-cell activation, including MHC class-I, CD86, CD40, and 4-1BBL [72,245]. Based on these observations, a French group has recently hypothesized that 4BL-derived B1 cells might induce a pathogenic, auto-immunogenic inflammatory response in MS [246]. Clinical trials aiming to characterize the phenotypic and functional properties of 4BL B cells in MS are currently underway (NCT03796611) [246].

In addition to a change in B1-cell frequency with advancing age, an alteration in B1-cell morphology and functionality was documented. Functionality-wise, beyond numbers, a reduction in the repertoire of B1 cells (i.e., in their B-cell receptor (BCR) diversity) was revealed [247,248]. Using single-cell sequencing analysis, Rodriguez-Zhurbenko’s group discovered a reduced IgM repertoire in B1 cells from aged compared to younger individuals [29]. The authors concluded that B1 cells display diminished diversity with advancing age and secrete fewer IgMs but relatively more high-affinity IgGs. The deterioration in B1-cell functionality in aging might impair the ability to counteract age-related neurodegeneration diseases [29]. In Alzheimer’s disease, for example, the capacity to opsonize and clear amyloid-beta aggregates by nAbs could be compromised [249]. Along the same line, it is conceivable that aging leads to a less-effective clearance of myelin debris near injured axons and contributes to MS progression. Figure 5 illustrates our putative model describing the age-related change in B1 cell repertoire in health and MS.

## 4. Interpretation 

This review examined the presumed involvement of B1 cells in MS. Unlike B2 cells, B1 cells are standalone, evolutionarily conserved, lymphoid-myeloid hybrid-cells with effector functions that span beyond first-line defense [45,250]. B1 cells’ ability to influence the adaptive arm of immunity is remarkable due to their continued innate memory development, self-renewal, and presumable residency in the human brain [89,251,252]. MS patients show fluctuating levels of B1 cells in the blood and CSF, observations that could link B1 cells to MS pathogenesis, either via a pro- or anti-inflammatory way [10,22,196,221,253]. Alteration in B1-cell equilibrium (specific subtype, frequency, or functionality) could dysregulate the immune response before or during MS relapse or trigger the disease progression and halt remission. As illustrated in Figure 4, B1 cells produce an abundance of protective, polyreactive, natural antibodies with specificities against myelin and other neuronal components. However, upon exposure to damaged myelin neoantigens, these nAbs can increase their specificity (i.e., pathogenicity) and become a “Trojan horse” inside the CNS, instigating a pro-inflammatory immune response and autoimmunity [109,185,254].

Why have B1 cells been ignored in the research of MS and other human neurodegenerative maladies? The main reason for the lack of sufficient translational evidence regarding the role of the human B1 cell in MS is perhaps the complex developmental path of B cells; B1-cell ontogeny is challenging to investigate in vivo, let alone inside the CNS. Additionally, as we have shown, B1-cell phenotypical and functional plasticity may camouflage its localization within tissues such as the brain. Once it has infiltrated the CNS and assumed its fully differentiated effector function, a B cell’s shift in surface markers might hinder its detection because it may be mistakenly recognized as another cell. Undoubtedly, one can argue that B1 cells differentiate and mature into B2 plasma cells or monocyte-like cells when reactivated inside the CNS. However, this possibility is questionable because B1 cells retain unique lymphocytic markers absent from B2 cells [22,49]. Additionally, one may speculate that upon activation and presence at the site of inflammation (e.g., meninges), B1 cells downregulate CD20 expression and become resistant to anti-CD20 depleting therapies. Indeed, it was shown that even though their number increases before relapse, they do not respond well to anti-CD20 (rituximab) treatment [255]. 

### 4.1. Emerging Therapeutic Strategies

#### 4.1.1. Cytokine Involvement in MS: A Double-Edged Sword

B1 cells are the predominant producers of IL-10 during the early phase of immune activation. IL-10 can be regarded in the context of the immune response as a “double-edged sword.” It has been established that MS patients show diminished IL-10 secretion by B1 cells (and Breg cells), suggesting that a defect in the B1-cell compartment could be involved in the disease [256,257]. Therefore, an exciting investigation line would be detailed profiling of the cytokines produced by B1 cells in active MS lesions or meningeal follicle aggregates. Complete profiling of the cytokines involved could lead to strategies to “replenish” dysregulated cytokine production (absence or excess thereof). However, an early phase II clinical trial testing IL-10 as a treatment in MS patients was terminated owing to a lack of effectiveness [258]. Systemically administered IL-10 could be quickly eliminated from the circulation or fail to reach sufficient concentrations in the brain. Nevertheless, the development of modern administration methods or gene therapies that would increase the stability and availability of IL-10 (or other cytokines) may prove to have better efficacy. Another cytokine constitutively produced by B1 cells that was tested is the tumor growth factor-beta (TGF-β). It appears that systemic administration of TGF-β ameliorated EAE, but, unfortunately, toxic renal effects were encountered [258].

#### 4.1.2. Manipulating B1-Cell Frequency and nAbs Properties

As discussed, the B1-cell phenotype and effector mechanisms shift throughout life and are significantly influenced by age, gender, and environmental factors. For instance, the increase in somatic hypermutations and enhanced affinity-maturation of IgMs (or class-switched autoreactive IgGs) could prime a more potent T-cell activation. Therefore, an approach targeting the restoration of the quantity and the quality of the B1-cell pool may confer novel therapeutic avenues. In addition, advances in stem-cell and genetic–immunotherapies could offer a remedy by replacing “old” B1 cell pools and restoring B1-cell immunoregulative functions.

A strategy of nAbs-replacement therapy is another emerging therapeutic opportunity that might prevent relapses and progression in multiple diseases [180]. In the case of decreased B1 counts in healthy elderly or MS patients, passive vaccination with nAbs against myelin neo-antigens may offer a therapeutic opportunity worth exploring. For example, administration of anti-myelin rHIgM22 nAbs could yield an advantage, as these IgM aAbs were shown recently to cross the BBB (in contrast to a previous conviction that their molecular mass and conformation prevent their influx into the CNS) [180]. Alternatively, intrathecal injection of a high dose of rHIgM22 into the CSF or transcranial administration into the meninges could be a possibility to promote myelin-debris clearance and to enhance remyelination [187,188,191,192,227,228,259]. Indeed, several studies in mouse models of MS have shown that rHIgM22 could enter the spinal cord and colocalize with demyelinating lesions [188,228,259]. Some authors proposed that rHIgM22 might promote remyelination directly by binding to cells in the lesions, leading to a reduction of the lesion load [225]. It was postulated that these antibodies directly recognize membrane glycolipids on oligodendrocytes. Whether the same effect could happen in humans is fascinating (though challenging to study in vivo). Assumingly, the first studies of rHIgM22 administration to RRMS patients in remission appeared to be safe, and no side-effects in phase I clinical trials were detected (trial IDs: NCT01803867 [229]; NCT02398461 [9]).

## 5. Conclusions

B1 cells participate in the body’s response to inflammation in several ways. They are present in the circulation and in tissue reservoirs, including the brain. They are positively selected to recognize self-antigens and to secrete immunoglobulins without the need for prior foreign antigen exposure. Furthermore, they can adjust their potency and improve their affinity to a specific (auto) antigen. This may equip them with innate memory powers, yielding a faster and stronger immune activation upon secondary encounters. Moreover, they are long-living and sustain self-renewal in the tissues (including the brain in mice). They can differentiate into phagocyte-like cells and secrete an array of inflammatory-mediating cytokines. Finally, they seem to be directly involved in myelination in the (developing) brain and may be involved in remyelination by producing nAbs against unidentified myelin antigen(s). B1-cell numbers and functional properties change before and during MS relapse and with advancing age. Building on the established concept that natural antibodies bridge the gap between innate and adaptive immune reactions [165], we propose that B1 cells form an “immune-response bridge” between both immune compartments (Figure 6). We speculate that dysregulation in the B1-cell compartment might disturb the integrity and elasticity of the immune-response bridge and contribute to the pathogenesis of MS. However, further studies are required to characterize the human B1-cell functional profile in MS. Deciphering its involvement may offer new opportunities for novel, more-effective therapies for MS and other neurodegenerative disorders. Figure 7 highlights the main conclusions of this paper.

## Figures and Tables

**Figure 1 biomedicines-10-00606-f001:**
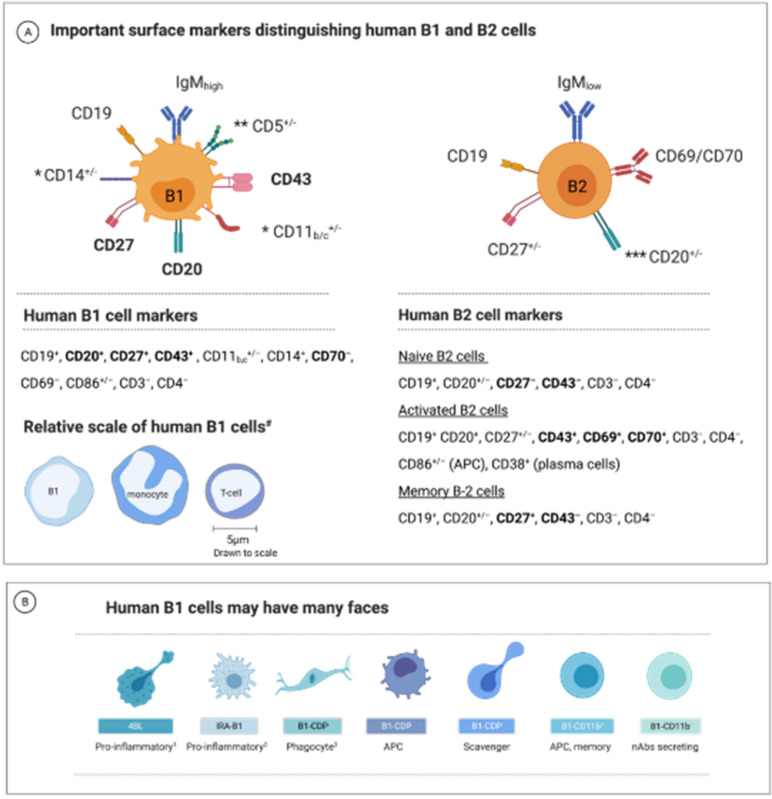
Currently proposed human B1 cell markers. (**A**) The human B1 lymphocyte population exhibits phenotypical, morphological, and functional diversity. The primary surface markers of the human B1 and B2 cells. In bold: the most prominent markers distinguishing human B1 and B2 (“conventional”) cells. Notes: * The integrin CD11b (CD18) expression is seen in about 10 to 15% of the B1 human cells. CD11c and CD14 are other myeloid-related markers expressed on some B1 cells. ** In mice, CD5^+^ B1 cells are termed B1a cells, whereas CD5^−^ B cells are B1b. This classification has not yet been confirmed in humans. The CD5 marker is also expressed on some T- and B2-cell subsets and is absent on some B1 cells. *** The expression of the CD20 marker depends on the differentiation state; B2 cells typically lose CD20 expression as they mature to plasmablasts but retain CD20 expression on memory cells. Bottom: The relative size of human B1 compared to monocyte and a T cell [52]. (**B**) B1 cells may have many faces—a schematic illustration of suggested differentiation state morphologies of human B1 cells (based on observations in mice or humans as described in the text; not to scale or exact morphology). The key functional features of each subtype are denoted. APC, antigen-presenting cells; ASC, antibodies-secreting cells; Notes: ^1^ 4BL cells are a B1-cell-derived subtype accumulating in aging and known to enhance CD8+ T cell production [65,72]. ^2^ IRA-B1 cell is a GM-CSF-secreting, pro-inflammatory B1 cell and polarizes CD4 T cells towards a Th1 profile [73]. ^3^ B1-CDP migrates to inflammatory sites and can phagocytose debris while maintaining lymphoid cell-surface markers [53]. Additional references discussing the human B1-cell surface markers: [21,23,25,52,74,75,76,77,78,79].

**Figure 2 biomedicines-10-00606-f002:**
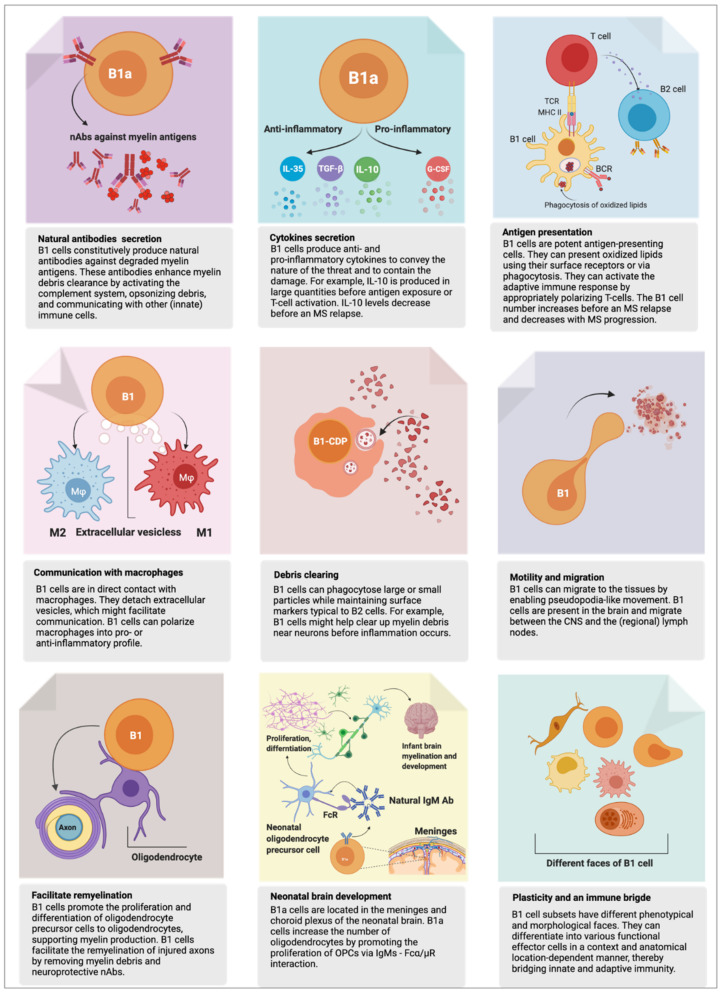
An overview of B1 cell functions. The B1 cell fulfills crucial homeostatic functions. However, it can act as a “double-edged sword,” playing a protective *or* harmful role. Here, we present an up-to-date overview of B1 cell function in health and possibly in multiple sclerosis (MS). The functions described here are based on experiments in mice or humans, referenced and discussed in detail in the text. Note: mouse B1a cells were recently shown to be present in the central nervous system (CNS), to facilitate differentiation of oligodendrocytes, and to support remyelination of injured axons (bottom left and middle panels). Whether this observation also holds to human B1 cells warrants investigation.

**Figure 3 biomedicines-10-00606-f003:**
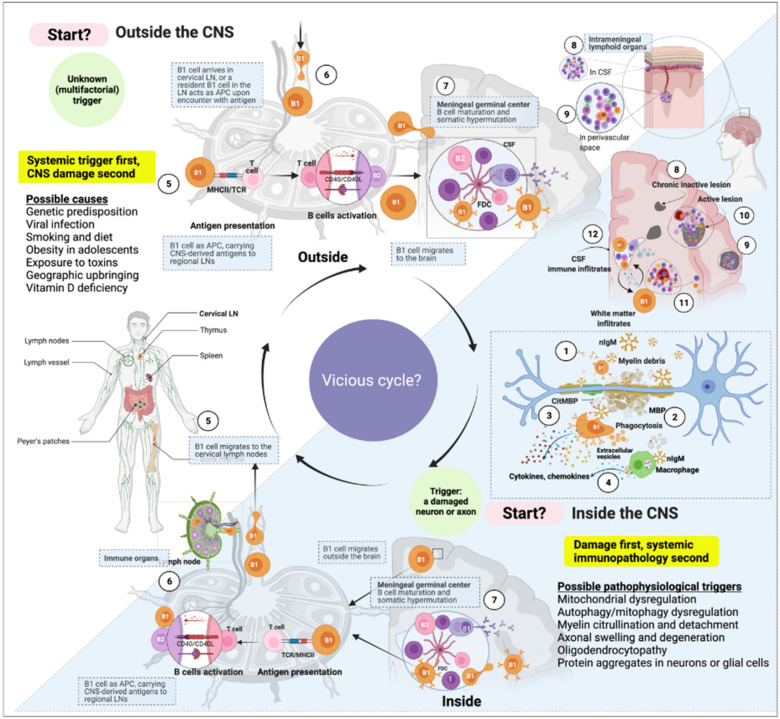
A cell’s journey according to the “inside-out” vs. “outside-in” paradigms of multiple sclerosis (MS). The immune response in MS might start at any point within the immune system or the central nervous system. The “inside-out” paradigm hints at an incidental instigation of (auto)immune response in the central nervous system (CNS) draining cervical and lumbar lymph nodes. According to the “inside-out” paradigm, this immune response activation is presumably caused by intrinsic damage to neurons or glial cells. The “outside-in” paradigm, on the other hand, represents an arbitrary starting point of the (auto)immune triggering from outside the CNS. In both scenarios, the immune response’s dynamic is cyclic and bidirectional; the glymphatic system might facilitate the influx and outflux of immune cells to and from the CNS. In this view, we propose that the immune system and central nervous system are fully integrated, where the CNS can virtually act as a (tertiary) lymphoid organ and the immune system as a relay station of systemic inflammatory signals. (1) B1 cells constitutively secrete natural antibodies (nAbs). The nAbs of the IgM class form high-avidity pentamers and attach to myelin on (or near) damaged axons. (2) B1 cell is a potent phagocyte that can internalize relatively large (insoluble) particles such as oxidized lipids aggregates and microbes (citrullinated myelin basic proteins (citMBP) is illustrated as an example). (3) Upon encounter with citMBP, B1 cells increase cytokine and chemokine production and attract other immune cells to the site of damage. (4) In addition, extracellular vesicles are often seen in the junction between macrophages and B1 cells; IgM attachment triggers inflammatory cytokine secretion by macrophages and enhances phagocytosis. (5) B1 cells loaded with antigens can migrate to the regional lymph node (e.g., the cervical lymph node) and present the captured antigen to T helper cells. (6) Follicular helper T cells activate B cells (either B1 or B2 cells) in the lymph node. (7–9) B1 cells can undergo affinity maturation and class-switching in germinal centers (inside lymph nodes or tertiary lymphoid organs in the CNS) and migrate back and forth to the brain or periphery. (10) depicted is an active lesion in the parenchyma’s grey and white matter. (11) Migratory white-matter immune cells. (12) Immune cells within the CSF in the lateral ventricle.

**Figure 4 biomedicines-10-00606-f004:**
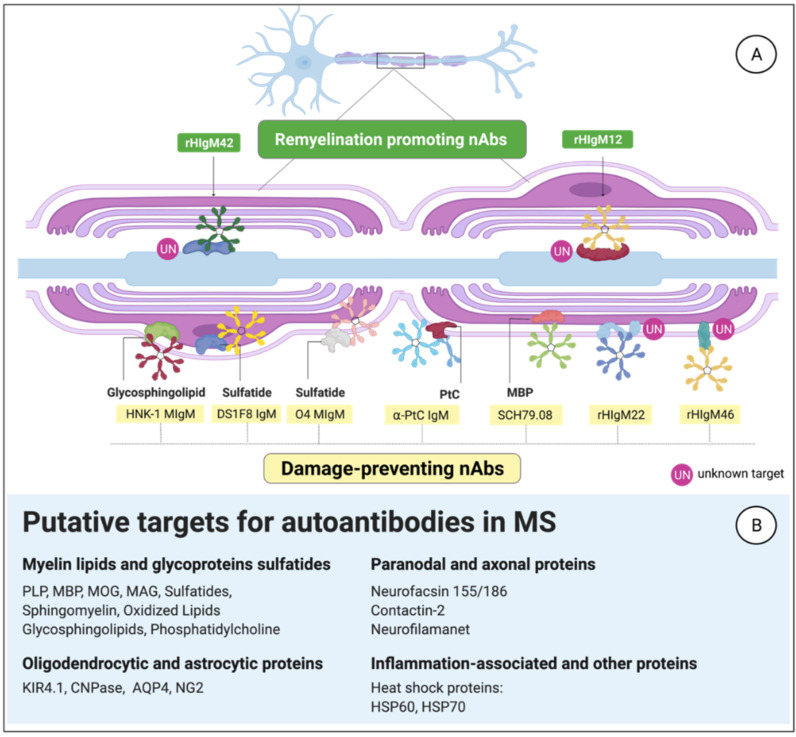
Natural antibodies evolved to fulfill homeostatic functions in the central nervous system (CNS). (**A**) Examples of IgM that can bind myelin or axonal antigens to exert housekeeping functions. (**B**) Putative antigenic candidates in the CNS may play a role in multiple sclerosis (MS).

**Figure 5 biomedicines-10-00606-f005:**
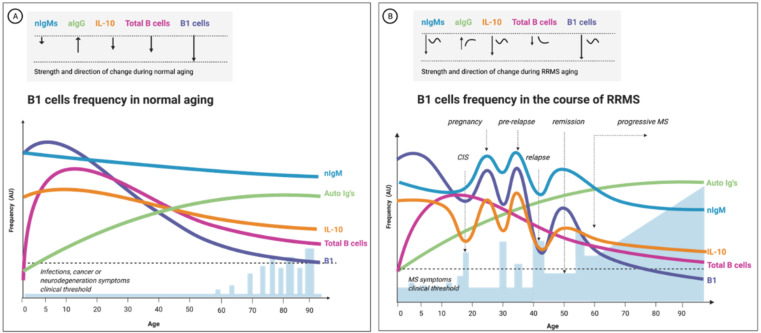
The change in B1-cell frequency during normal aging and remitting-relapsing multiple sclerosis (RRMS)-proposed model based on current empirical evidence. B1-cell frequency is positively correlated to the number of natural antibodies and inversely correlated to the levels of IL-10 in the blood. In (**A**), the frequency has a linear trend: the B1-cell count declines with aging proportionally to the total B-cell population (all B cells subtypes). (**B**) shows the expected change in the frequency of B1 cells during aging of a female RRMS patient; here, the fluctuations are more accentuated, the onset time of the change differs, and the trend is often nonlinear. Moreover, the change in B1-cell frequency and serum IL-10 correlate to the clinical picture (B1-cell numbers increase before an attack and decline thereafter): nIgM, natural IgMs antibodies; aIgs, autoreactive antibodies (mostly class-switched IgGs); total B cells, the total B lymphocytes and B1-cell- frequency in the peripheral blood circulation; CIS, clinically isolated syndrome; blue bars represent relapse episodes; the dashed line depicts the threshold for clinical symptoms manifestations; AU, arbitrary units.

**Figure 6 biomedicines-10-00606-f006:**
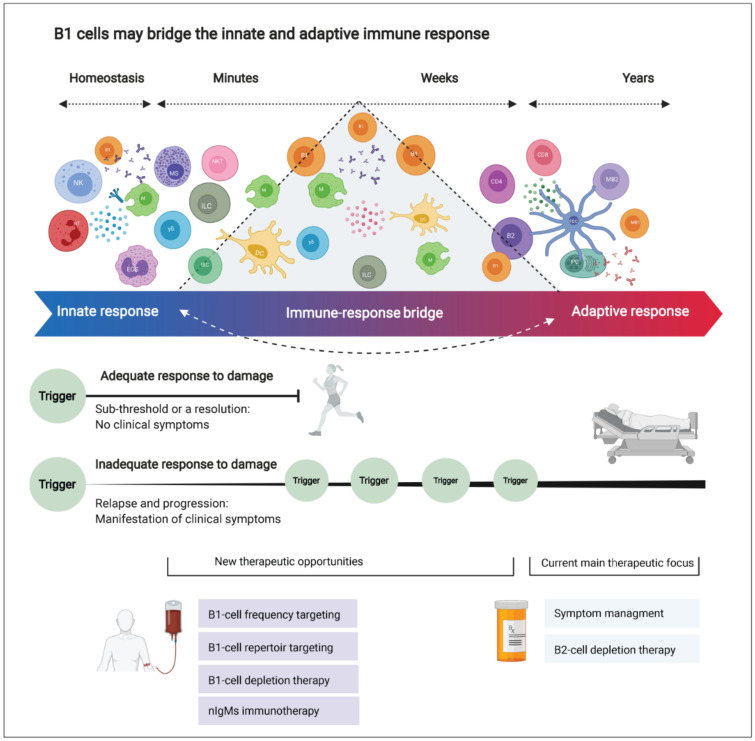
B1 cells may bridge the innate and adaptive immune response, and future therapeutic strategies for multiple sclerosis (MS). Innate immune cells are distributed in the brain to react against damage rapidly. On the other hand, adaptive immune cells reside in lymph nodes (mainly outside the central nervous system; CNS) and function as highly specialized cells providing antigen-specific, robust, yet slow-to-react immune response. Here, we question this binary division as new evidence points out the blurred lines between the innate and adaptive immune response. The immune response is a continuous, bidirectional, elastic spectrum of defense mechanisms rather than a binary division into “innate” or “adaptive” immune responses. This elasticity might be impaired in MS (and possibly in other CNS-degenerative diseases) due to dysregulation or aberrant functioning of specialized “bridging” cells such as the B1 cell. In aging and MS, the dysregulated B1-cell compartment can disturb the delicate balance between damage and repair in the CNS.

**Figure 7 biomedicines-10-00606-f007:**
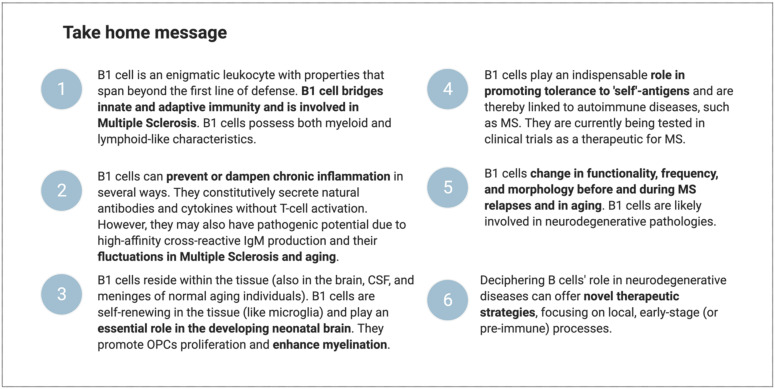
Take home message.

## Data Availability

Not applicable.

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
