# Peer review of "The Forgotten Brother: The Innate-like B1 Cell in Multiple Sclerosis"

_biomedicines, 2022, doi:10.3390/biomedicines10030606_

Round 1

Reviewer 1 Report

Current review article introduced the role of 'innate-like' B1 lymphocyte in multiple sclerosis (MS). Please conduct the concerns below.

  1. This report introduced the functional role of B-lymphocytes in a good way. However, B1 cells secrete more IL-10 than B2 cells that was in brain or in periphery? Please mention in clear.
  2. B1 cells also produce pro-inflammatory cytokines that was observed in CNS or serum?
  3. The sentence indicating “in individuals infected with SARS-CoV-2 virus (B1a cells)” from Reference 163 and 164 seems unclear. Is it means that SARS-CoV-2 virus as B1a cells? Please explain it in detail.
  4. In Figure 2, functions of B1 cells were showed in very clear. However, the part whether it is the case also with human B1a-like cells seems better to mention in the legends.
  5. The anti-myelin rHIgM22 nAbs seems useful in the future. Please more information before conclusion.
  6. The dysregulations in the B1 cells are responsible for MS only? Or it may also associate to another neurodegeneration?

Overall, this is an interesting and useful review article. The indicated figures will be widely cited by others in advance.

Reviewer 2 Report

In this manuscript titled “the Forgotten Brother. The Innate-Like B1 Cell in Multiple Sclerosis”, the authors reviewed the recent findings on the roles of B1 cells in multiple sclerosis. The authors started this review with an introduction of multiple sclerosis and B1 cells in general. Then the authors reviewed different aspects of B1 cells in multiple sclerosis such as their contribution to the autoimmune response in MS, their production of autoreactive antibodies and their correlation with MS progression. Overall, this is a nicely organized and well written manuscript. However, I think several points, especially some conceptual confusion, need to be clarified before it reached the quality for publication.

Major issues:

  1. From this review, it seems that B1 cells can secrete both pro-inflammatory cytokines and regulatory cytokines and they can be both protective and pathological in MS. The authors didn’t spend enough effort to elaborate the specific context determining the pathological vs protective roles of B1 cells in MS. As an example, page 9, “On the same line of evidence, eliminating B1 cells in mouse models reduced autoimmunity”, is there any evidence that eliminating B1 cells in mouse models increased autoimmunity or pathology in MS? If there is, under what context?

  1. The authors didn’t spend enough effort to distinguish the results from human vs the results from mice, especially in the figures. For example, figure 1 is mostly about the subsets of B1 cells in human, the only note related with mice is “** In mice, CD5+ B1 cells are B1a cells, whereas CD5- B cells are B1b”. Then, can I assume that mice have all the counterparts of these human B1 cell subsets in figure 1 besides this note? Another example, nearly all the studies in the section “Humoral response – arrows to the target” seem to be from mice. Is this the case? If not, please clarify. Figure 2 is very confusing because there is no annotation of whether these results are from human or mice.

  1. In the abstract, “This review focuses on a relatively little explored innate immune cell — the 'innate-like' B1 lymphocyte”, the authors may want to rephrase this sentence (such as deleting the first “innate”), because it is still debatable whether B1 cells are innate immune cells or innate-like adaptive immune cells. As the authors mentioned, “Upon activation, B1b cells migrate to secondary lymph organs and differentiate into Abs producing plasma cells. They undergo somatic hypermutation and affinity maturation, increasing their specificity (page 5)”, this is the typical characteristics of adaptive immune cells.

  1. It will be nice if the authors can describe the research findings more quantitively. For example, page 2, “B1 cells constitute a small percentage of the adult leukocytes pool in adult humans”, what is the approximate range of the percentage? In section “3.4. B1 cells frequency correlates with relapse and MS progression” and section “3.5. MS progression is age-related; so is B1 cells' quality and frequency”, the authors mentioned the frequency of B1 cells changed in several circumstances, what is the magnitude of the changes? From what percentage to what percentage? How many folds?

Round 2

Reviewer 2 Report

All concerns resolved